# Limitations to sustainable renewable jet fuels production attributed to cost than energy-water-food resource availability

Cheng Tung Chong [1] ✉ & Jo-Han Ng [2] ✉

Renewable jet fuel (RJF) is often touted as the only viable sustainable energy source for the aviation sector, given the difficulties faced by other low-carbon energy sources in overcoming technological barriers. Despite that, the sustainability of RJF is still in dispute due to the conflicting requirements in natural resource for producing the fuels. We introduce a holistic 25-indicator sustainability index encompassing the four domains of energy-water-food nexus and governance, that measures the potential impact of RJF production on 154 countries (and territories) through the oil-to-jet, alcohol-to-jet and gas-to-jet conversion methods. Countries and territories are ranked according to the composite index scores of the four domains. The sustainability index model provides insights on how RJF affords the aviation sector a clean slate in determining the manner of development in a sustainably and equitable way, while also marching towards the long-term goal of carbon neutrality, in alignment with the Sustainable Development Goals.

The aviation sector contributes ~2.5% to the global $CO_2$ emissions[1]. Taking into consideration other non-$CO_2$ climate forcers emitted by aircraft such as nitrogen oxides, aerosols, ozone precursors, contrail cirrus, the net radiative forcing impact on climatic warming is about 3.5%[2,3]. By 2050, the consumption of jet fuel is expected to reach 230 billion gallons, a two-fold increase from the present 106 billion gallons[4], thus the growth of the industry will inevitably result in the rise in greenhouse gas (GHG) emissions. To decarbonise the sector, the International Civil Aviation Organization (ICAO) has proposed a roadmap to reduce the carbon footprint. Among the measures, renewable jet fuel (RJF) has been identified as the primary method for aviation decarbonisation[5]. RJF that meets the metric of sustainability, i.e. lower carbon footprint than conventional jet fuel, is known as Sustainable Aviation Fuels (SAF). ICAO has published a list of verified low carbon SAF, known as the CORSIA Eligible Fuel (CEF), that can be used by the airline operators to achieve carbon emission reductions[6]. CORSIA stands for Carbon Offsetting and Reduction Scheme for International Aviation, which represents a set of global cooperative measure by ICAO member states to reduce emissions for international aviation while minimising market distortion. The CEF is ascertained by

the working group under CORSIA using the key life cycle assessment (LCA) value, which is calculated by accounting the feedstock cultivation, land-use change emissions, direct and indirect energy and material requirements to ascertain the total global warming potential impact[7]. Such LCA values provide the basis for calculating the GHG savings achieved when using CEF.

To date, a total of seven RJF production pathways have been certified by the American Society for Testing and Materials (ASTM D7566 - Standard specification for aviation turbine fuel containing synthesised hydrocarbons) since 2009[8]. The certification is important to enable the application of RJF in actual aircraft, ensuring the blend of the RJF with conventional jet fuel meets the specifications required for safe flight operation. Presently, the RJF production pathways can broadly be categorised as OTJ (oil-to-jet), AJT (alcohol-to-jet) and GTJ (gas-to-jet)[9]. More emerging production pathways are expected to be certified in the near future, as significant progress in RJF technology has been achieved in recent years using different conversion processes[4]. The recent crude oil price spike to about 130 United States Dollar (USD) per barrel[10] (March 2022) due to the Ukraine-Russia conflict has severe implication to the cost of RJF production. The

[1]China-UK Low Carbon College, Shanghai Jiao Tong University, 201306 Lingang, Shanghai, China. [2]Carbon Neutrality Research Group, University of Southampton Malaysia, 79100 Iskandar Puteri Johor, Malaysia. ✉e-mail: ctchong@sjtu.edu.cn; J.Ng@soton.ac.uk

economics of RJF production is favoured at high crude oil price as the asking-to-selling price gap is narrowed, but the stress on local resources such as feedstock may result in higher environmental cost.

To ensure the sustainability of RJF production at large scale, a wider spectrum of metrics to better capture the spatial-temporal heterogeneity of the climate forcing agents and impact of RJF production need to be considered in tandem[11]. The life cycle $CO_2$ emissions value[7], which is the primary method used in the industry to quantify the climatic impact, is insufficient to ascertain environmental sustainability. The use of bioresources for RJF production is closely interlinked with the local resources such as food, water, land, energy and forests, as well as with economy and society. A paradigm shift via a multi-factor approach is vital in assessing the RJF development needs[12], by aligning the technological transition with the environmental and socio-economical aspects to ensure long-term sustainability.

The Sustainable Development Goals (SDGs) framework consisting of 17 goals, 169 targets and 147 indicators is adopted by the United Nations with the aim to stimulate action plans of critical importance to the people, planet and prosperity[13]. The SDGs have become the guiding principles for sustainable development plans by considering the balance of economics, social and environmental. While the SDGs emphasise on the equitability of social and natural resource sustainability, the indicators better represent socio-economic development rather than biodiversity conservation[14]. This might lead to a scenario where countries or territories are meeting their environmental SDG commitments while still causing substantial destruction to the environment. As such, alternative measures of sustainability which is complementary to the SDGs are required to resolve the conundrum between climate change mitigation and the SDGs. It is pivotal that they are purpose-built for each industry.

In the present work, we conduct a data-driven, techno-econometric analysis that incorporates elements of top-down and bottom-up modelling approaches to quantitatively rank countries and territories in an energy-water-food (EWF) nexus plus governance-based biojet fuel sustainability index (see "Methods" section). The inclusion of governance as the fourth pillar and economic criteria counters the criticisms levelled at the EWF nexus approach which often ignores the well-being and basic subsistence of people. This approach balances the aggregation at top-down macro-level for behavioural realism while disaggregating from bottom-up micro-level to accurately represent the technological potential. Our analysis covers 154 countries and territories meeting the minimum data threshold requirement, with nine crude oil price and biojet fuel production method permutations. We obtain the biojet fuels sustainability index scores for each country and territory from 25 indicators as shown in Fig. 1, then ranking them by the EWF securities and governance domains, before aggregating them into an overall ranking.

Here, we report in the following order, covering the global biojet fuel sustainability index by ranking the countries and territories on their sustainability measures, followed by the three key aspects of the nexus, such as energy replaceability, water footprint, and food along with feedstock. Then, we discuss the economics aspects on the profitability price point and the overall limiting factors within the resource nexus. Next, we elaborate on the sustainability of RJF from the SDG perspective. With this, the study aims to advocate for the use of a sustainability index for RJF to complement the pursuit of the SDGs, while enabling a more comprehensive assessment of RJF sustainability tailored to the aviation sector. This provides insights to support the aviation industry's efforts to meet the global sustainability targets.

## Results
### Global biojet fuel sustainability index
The COVID-19 pandemic has irrevocably changed the global energy market, with the largest ever price swings between trough (March 2020) and peak (March 2022) in crude oil price within a two-year period being recorded at around -USD 37 and USD 130 per barrel, respectively[10]. While the former is due to logistic and petroleum storage issues, and the latter is a consequence of supply fears, the worldwide inflationary impacts of higher energy costs are expected to push crude oil prices eminently upwards.

We set the key scenario for the crude oil price of USD 135 per barrel, with biojet fuel production costs of OTJ, ETJ and GTJ at -USD 0.6378, USD 0.6334 and -USD 0.1824 per litre, respectively. In this scenario, Finland emerged as the top-ranked country with a sustainability index score (SIS) of 57.4 out of a possible 100 as shown in Fig. 2. See Supplementary Dataset 1 for the full ranking and index scores. This high SIS stemmed from a thriving RJF industry to rank first in the Energy domain, having robust governance to rank 4th, good food security with sufficient production to withstand shocks and being invulnerable to food shortage due to easy access to the food market. Overall, the country emerged in the top quartile for the energy, food and governance domains, while still having above-average water security.

Like Finland, most of the other countries in the top 10 have the distinction of being in the best quartile for 2-3 of the sustainability domains. There is no country or territory which is outstanding on all four domains as energy is heavily dependent on the availability of low-cost feedstocks, water security relies on having not already tapped into their water resources for other industries, food security is determined by either having easy access to food market or already producing surplus food, while good governance is required for policies to be passed through strong political will. This reinforces the idea that even country presently with the best sustainability rating will find it difficult to balance the utilitarian aims of using the finite EWF resources for sustainable aviation.

European countries dominate the sustainability ranking with 13 countries in the top 30, with the general characteristics of having secured food supplies and functioning governments, while being merely above average in water security but having low scores in energy security. The mostly South American countries in the Americas form the next largest group with nine being in a good position to push for RJF productions. Unlike the distinct characteristics of the European countries, the Americas countries (or territories) are a disparate group, each with their own strengths and weaknesses. Asia-Pacific countries with three apiece from Asia and Oceania round up the most sustainable countries for RJFs, with one coming from Africa. The African countries are generally ranked low for energy security due to issues in not having affordable feedstocks for RJF production and are still highly dependent on energy imports.

For individual domains, the energy security ranking favours countries with larger land area such as India, Brazil, USA and Indonesia as there are likelihood of more biomass resources that can potentially be collected for RJF productions, whether from crops, forest or agricultural wastes. Countries highly dependent on seed oil exports like Ukraine, Indonesia and Malaysia also tend to do well as there are abundant feedstock that can be converted to RJF using OTJ methods. Water security is tied to economic activities, so countries with additional allowances to divert water for RJF productions are also those with minor agricultural sector or minimal water-intensive industries, such as Somalia, Niger, Suriname, Mongolia and Gabon. Countries with high food security scores, in particular those in the top quartile, are usually wealthy countries with access to a well-regulated food market such as Switzerland, Belgium, Netherlands and Norway. Food security can be fragile even for highly ranked countries like Lebanon. The Russia-Ukraine conflict has severely jeopardised food security for Lebanon as it is reliant on imported food particularly sunflower oil from Ukraine[15]. The high-income countries as classified by the World Bank from Scandinavia, Western European and Australasia such as Norway, Sweden, Finland, Denmark, Netherlands, Switzerland, Luxembourg, New Zealand and Australia usually have high government effectiveness, leading to high placing in governance ranking.

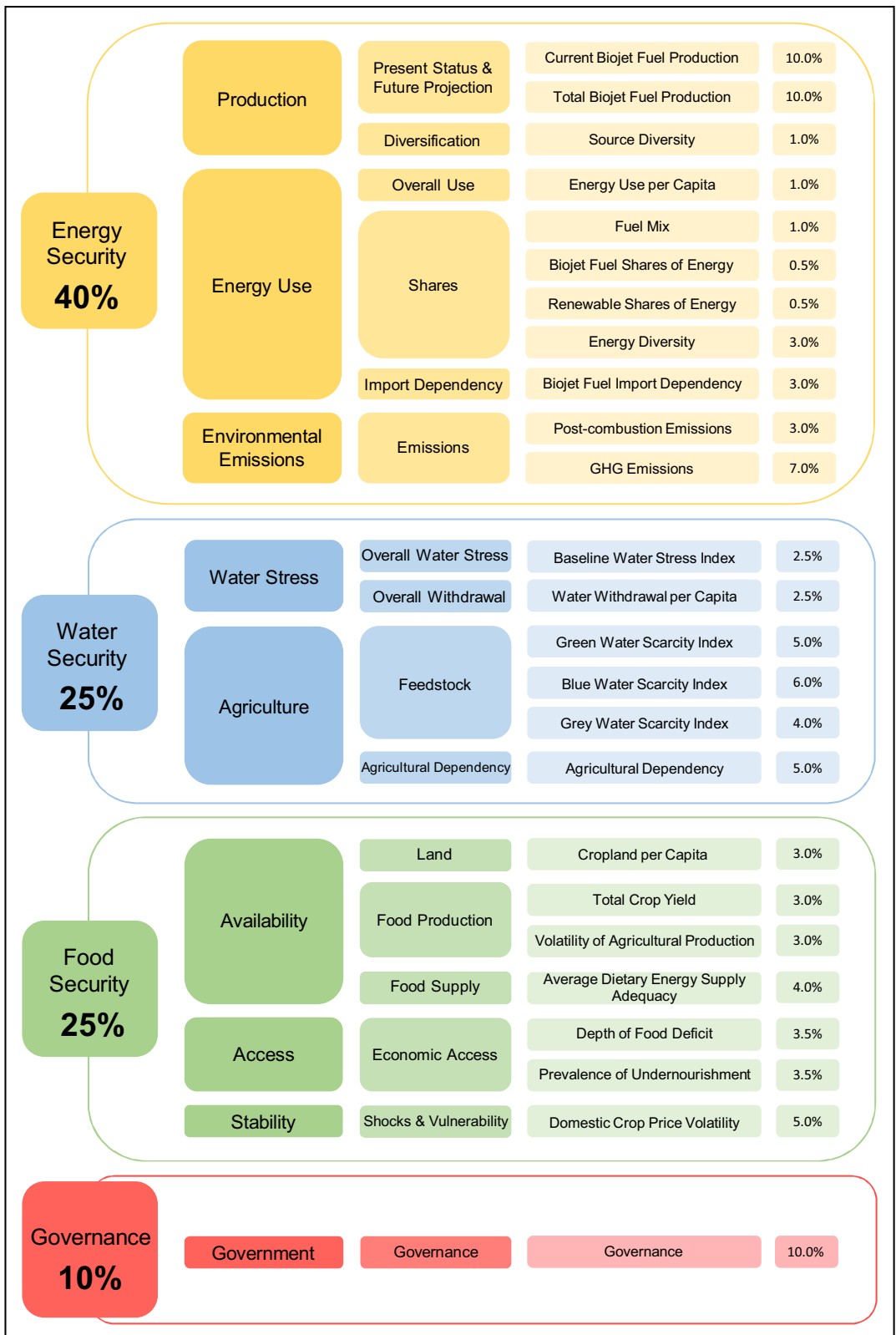

**Fig. 1 | Sustainability index indicators.** The categories of the global biojet fuel sustainability index covering the finite resources of energy, water and food, plus governance. The four main categories are subdivided into nine themes, 16 subthemes and 25 indicators.

### Energy replaceability

Energy replaceability refers to the percentage blend that a country or territory has to displace existing conventional jet fuel usage. Presently, the ASTM-approved technological pathways for RJF only allows blending level up to 50% by volume. The limits were triggered by the lack of aromatics in RJF, which are required for high altitude usage, as all mixtures of RJF and conventional jet fuel must be undifferentiated once they are mixed, also referred as drop-in fuel. The RJF energy replaceability levels for each location when crude oil price is at USD 135 per barrel and there are provisions for partial subsidies to offset

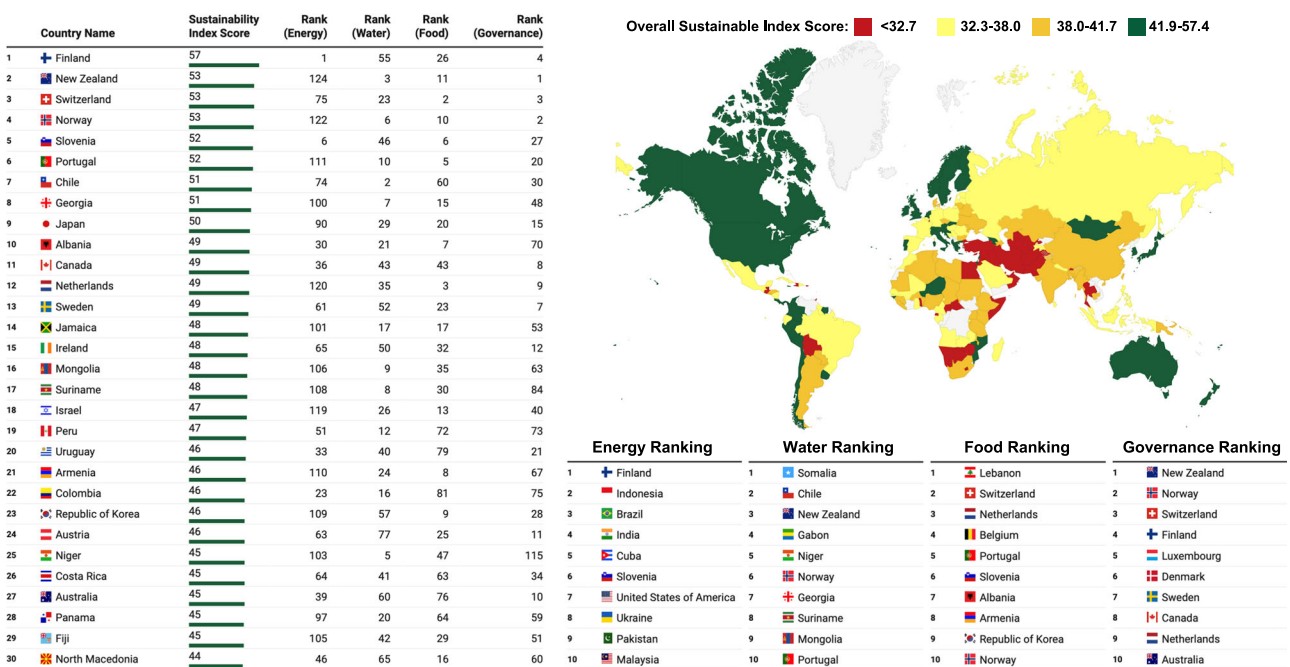

**Fig. 2 | Global biojet fuel sustainability index ranking.** The sustainability index scores for each country or territory is aggregated from the individual domains of energy, water, food and governance ranking. The colours in the choropleth map are categorically arranged in descending quartile order of green, orange, yellow and red.

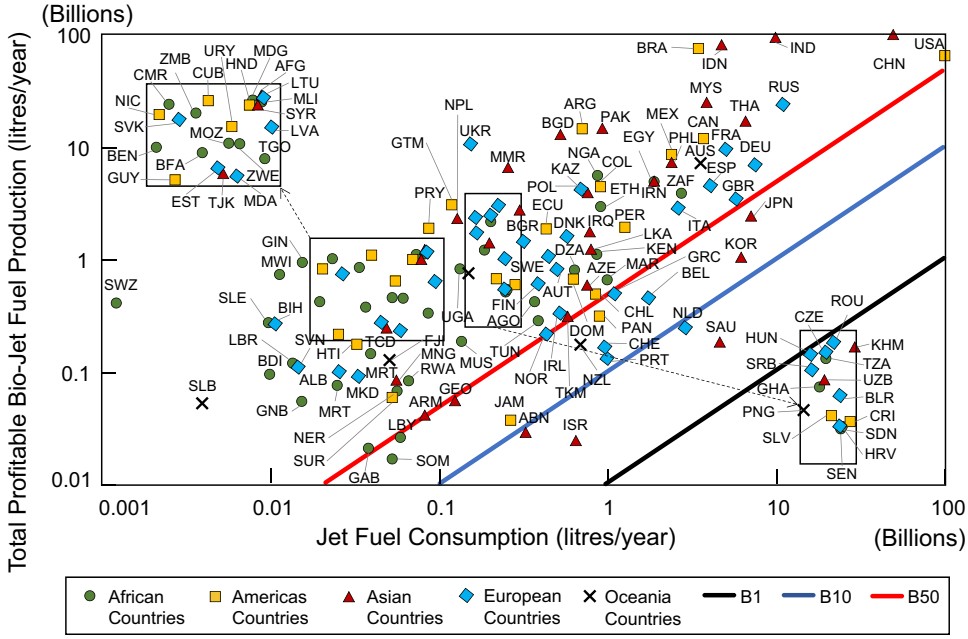

**Fig. 3 | Replaceable jet fuel consumption level by renewable jet fuel (RJF).** The total potential profitable RJF quantity that can replace present day petroleum-based jet fuel consumption at crude oil price of United States Dollar (USD) 135 per barrel with partial subsidies to defray production costs. The B1, B10 and B50 lines refer to 1%, 10% and 50% biojet fuel in the volumetric blend with fossil jet fuel, respectively. As most RJFs can only be mixed at 50% volumetric level to be classified as a drop-in fuel, countries above the diagonal red line have the potential to maximise the usage of RJF as a mixture component in a profitable manner. Insets are used to provide space for labelling. The abbreviation of the listed countries follows the ISO 3166-1 by the International Organization for Standardization (ISO), and is provided in the Supplementary Dataset 1.

production costs, as shown in Fig. 3. It should be noted that the replaceability level worldwide is near identical even at USD 60 per barrel crude oil price, as the main deciding factor is the provision of financial support by governments. The support for this nascent industry is required until critical mass is reached and know-how in production technologies improves to drive down the production costs. Once the economies of scale are achieved, governmental subsidies can be removed.

As RJFs can be produced from a myriad of feedstocks, 74% of the 154 countries and territories analysed in this study can maximise the

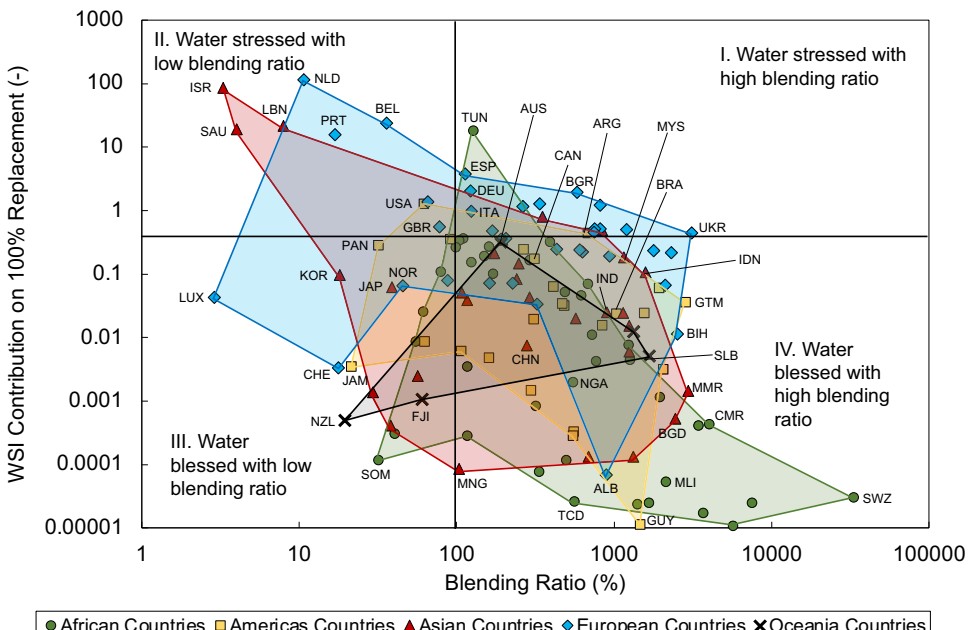

**Fig. 4 | Projected water stress index (WSI) against the potential blending ratio.** Projected WSI on 100% replacement against the potential blending ratios as banded by continents. The four quadrants represent water stressed-blessed and high-low blending ratio dichotomies. A WSI value of 0.4 is classified as water stressed, while values above unity are only hypothetically plausible. Countries in the same continent are linked with lines. The abbreviation of the listed countries follows the ISO 3166-1 by the International Organization for Standardization (ISO), and is provided in the Supplementary Dataset 1.

blending levels of RJF with conventional jet fuels in a profitable manner under favourable conditions, often with huge surpluses. Notably, few of the biggest jet fuel consumers, namely the USA, China, Russia, India and Brazil which collectively utilise 52.4% of global jet fuel, can exceed the maximum RJF blending levels of 50%. Japan, South Korea, Saudi Arabia, Netherlands and Belgium are among countries with high jet fuel consumption above 1 billion litres annually but are unable to achieve maximum blending levels (positioned below the red diagonal line) even under favourable economic conditions.

On the other hand, there are 13, 4 and 21 countries with the potential capacity to volumetrically displace only 10-50%, 1-10% and sub-1% of fossil jet fuels, respectively. Majority of the countries and territories with limited capacity to ramp up their RJF production are the countries with small land mass and limited agricultural output like Singapore, Malta and Luxembourg, or Middle East and North Africa (MENA) countries with generally lower land productivity such as Bahrain, Jordan, Kuwait, Oman, Qatar and United Arab Emirates, and are already managing food insecurity[16].

## Water footprint
Using a four-quadrant system in Fig. 4 to evaluate the balance between displacing conventional jet fuel with RJFs and water stress index (WSI), it is observed that every continent would have countries or territories that could develop their local RJF industries without overwhelming the water usage. If WSI above 0.4 is considered high and 100% blending ratio denotes total displacement of fossil fuel, then we have a water stress-blessed and low-high blending ratio matrix.

Every continent has countries or territories with the potential to totally replace fossil-based fuels with the more sustainable alternative without incurring water stressed scenario (i.e. in Quadrant IV). The continents of Africa, Asia, Americas, Europe and Oceania have 32, 20, 18, 14 and 3 countries, respectively. In general, countries or territories in Oceania, Americas and Africa are less likely to have stress on water security. This is particularly crucial for countries or territories in Oceania as the predominant method to access these countries or territories for passengers is via airways, so there is an added incentive to

improve energy security and be less susceptible to crude oil price volatility. Eighteen European countries will expect some degree of water stress (i.e. in Quadrants I and II) at 100% jet fuel replacement level.

## Food and feedstock
Food supply chain stresses do not only stem from the lack of production, but also contributed by bottlenecks in the processing, transport and logistics, as well as major shifts in demands. The Organization for Economic Co-operation and Development (OECD) found that the COVID-19 pandemic has introduced unexpected stresses on the food supply chains, although also noted on the resilience of the supply chain actors to repivot themselves to resume the availability of food[17]. Policy makers were found to have avoided the same errors committed during the 2007-8 food price crisis. As such, it will be more strategic to look at land usage in addition to food stress, as the former is permanent while the latter is transient.

The total plantation area needed across all RJF pathways to tap into the full potential of the fuel amounts to 1.508 billion hectares. This number is the same as the global amount used in crop production, as the simulation model does not assume increased agricultural output, but instead divert oil, sugar, starch and utilise waste feedstocks for the RJF production. The production methods of OTJ, ETJ and GTJ are expected to co-utilise 24.6%, 23.6% and 51.8% of plantation land on top of normal agricultural practices, respectively. This remains positive as the release of captured and sequestered carbon within fossil-based jet fuel through aviation-sector combustion can be reduced without increase in precious arable land usage. As the increase in RJF feedstock and food availability are tied to agricultural land availability, there is little to be concerned on the food vs fuel debate, as there remains 2.7 billion hectares of land that can be brought under cultivation as estimated by the Food Agriculture and Organization of the United Nations[18].

## Profitability price point
The OTJ, ETJ and GTJ family of production methods have the largest distinction, with very different feedstock and biomass quality

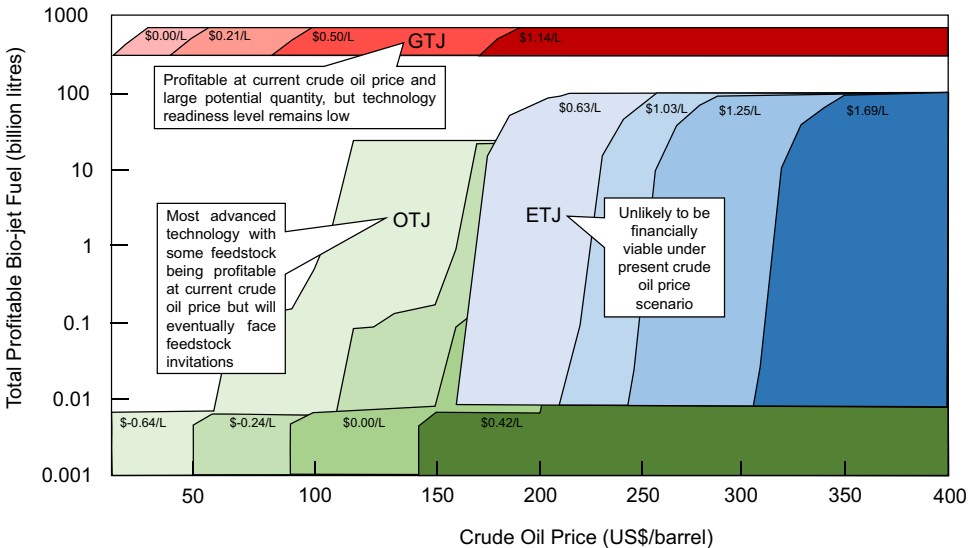

**Fig. 5 | Profitability price point of biojet fuels by production technologies.** Projected profitable biojet fuel volume for the oil-to-jet (OTJ), ethanol-to-jet (ETJ) and gas-to-jet (GTJ) production methods when crude oil price varies in the United States Dollars (USD) 0-400 per barrel range. The production method of OTJ, ETJ and GTJ is shown in shades of green, blue and red, respectively.

requirements. The OTJ method is best represented by the prevalent hydroprocessed esters and fatty acids (HEFA) which involves hydrotreatment, cracking and isomerisation of the feedstocks. It is also the most technologically advanced production method with a technology readiness level (TRL) among the ASTM-certified pathways at TRL 9. Despite the OTJ method allowing biojet fuel producers to turn in a profit at various crude oil price-production cost combinations as illustrated in Fig. 5, OTJ will eventually face feedstock availability limitations. There are only an estimated of 35.15 billion litres of feedstock that can be sustainably diverted from food sources for biojet fuels without impacting food supplies. Furthermore, some of the crude oil price-production cost permutations would require levels of governmental subsidies, pioneering incentives, and supportive policies to function.

Ethanol-to-jet, which is part of the alcohol-to-jet methodology, involves the relatively mature dehydration, oligomerization and hydrogenation processes. It is unlikely to be financially viable under the present market price for crude oil. The industry for ETJ is also unlikely to receive governmental subsidies as the amount will be prohibitively exorbitant due to the already existing and thriving bioethanol industry globally. In the event if crude oil price increases above USD 165 per barrel, ETJ will be immediately viable and also have a higher ceiling for feedstock availability at 94.04 billion litres.

The syngas-based GTJ method has the greatest feedstock availability, which is an order of magnitude higher than that of OTJ and ETJ. This method has sufficient feedstock to produce 704.32 billion litres of RJF, as any organic biomass has the potential to be turned into GTJ feedstock. As agricultural wastes and other non-edible feedstocks are earmarked for valorisation as GTJ biojet fuels, the feedstock cost is a fraction of OTJ and ETJ feedstocks. This means that GTJ method is economically viable even at present crude oil price with zero to minimal governmental assistance. The only drawback is the low TRL, as the jet fuel range selectivity of the dominant GTJ method, namely the Fischer-Tropsch (FT) method remains low at larger scale productions.

**Limiting factors**

The delicate balance between the resources nexus requirements and economic climate of the aviation fuel industry requires limiting factors to be determined. This is to avoid the depletion of any individual resource and provide economics insights to policy-makers on the immediate dangers of scaling up their RJF industry. The limiting factors to produce profitable RJF for each country or territory at USD 60 per barrel and USD 135 per barrel with various subsidy scenarios are shown in Fig. 6. The subsidy amounts are benchmarked against the various US biofuel policies. The five limiting factors include energy diversity, water stress, food stress, feedstock availability and crude oil price.

The limiting factors are generally insensitive towards crude oil price in the USD 60-135/barrel range. Instead, they are almost exclusively influenced by the availability of the hypothetical governmental subsidies. In a 'no subsidy' scenario regardless of crude oil price, close to 90% of all countries and territories will be limited by the prohibitive economic factor to produce profitable RJF.

However, even if 'partial subsidy' is provided, then primarily EWF nexus-type limiting factors surface as profitability conditions become less of a concern. This might lead to RJF becoming a victim of their own success where a glut of RJF will enter the aviation sector for 80 countries and territories, leading to concerns about a lack of diversity in fuel sources. Such a risk is pertinent for localities which are over-reliant on a smaller subset of feedstocks, for example, Indonesia might have sufficient palm oil for RJF production, but the 2022 palm oil export ban in the country shed light on the dangers of using a dominant crop as a feedstock. Food stress and feedstock availability would affect 40 and 24 countries and territories, respectively. They either have socio-economic issues such as those in the northern African region, or have large swathe of barren land like Mongolia and Australia. There are still 10 countries that struggle to compete with petroleum-based jet fuel even with generous subsidies pumped in due to high feedstock costs. In all scenarios, water stress is unlikely to be the limiting factor as it is often caused by multiple sectors rather than primarily caused by biofuels.

**Sustainability of RJF from SDG perspective**

The relationship between the RJF production with the social and natural resource sustainability is mapped via the SDG as shown in Fig. 7. The production of sustainable and affordable RJF is directly linked to SDG 7, which serves as the foundation that leads to the spillover effects on the broader scale of sustainable development. The most direct impact of RJF production is energy diversity, water and food, which are closely linked to the SDG 8, 6 and 1, respectively. Development of the RJF industry will lead to job creation and economic growth, in which

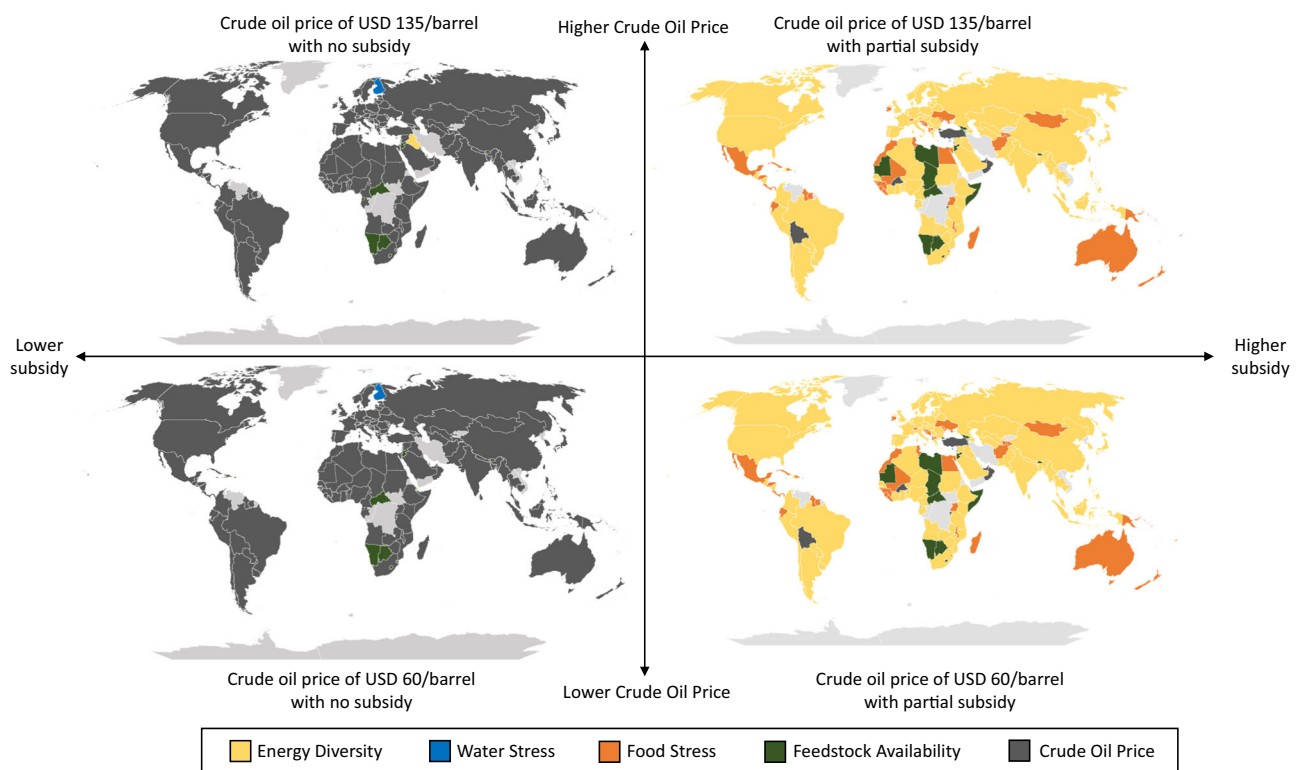

**Fig. 6 | Limiting factors for the potential profitable renewable jet fuel (RJF) production.** Limiting factors for each country and territory when crude oil price are in the range of United State Dollars (USD) 60-135/barrel under a binary subsidy scenarios. The colours represent the most critical limiting factor that countries and territories will face to produce profitable RJF.

the spillover effects include the innovation in industry and infrastructure, paving the way for the development of a more sustainable city. From the resource perspective, responsible consumption of water and feedstock for RJF production is essential to safeguard the health and well-being of society. The utilisation of sustainable RJF on aircraft leads to reduction of GHG emissions, which is a positive contribution to climate change that will cascade to life on ground and water, so that the livelihood of society is protected. In the post-pandemic era, it is expected that RJF demand will soar in the pursuit of long-term carbon neutrality goal. The alignment of sustainable RJF with the environmental and social-economy aspects is pivotal to ensure long-term sustainability.

## Discussion

Our analysis affirmed the plausibility of introducing conventional jet fuel-RJF blend as the de facto aviation sector fuel, by simulating the EWF nexus concerns and financial viability against various crude oil price and subsidy scenarios. However, the study also untangles the individual constraints of the EWF nexus alongside governance, and found that it will be unrealistic to expect all countries and territories to do well in the individual energy, water and food factors as present socio-economic advantages and geographical lottery play a large role in determining the suitability of a country or territory in producing RJF in a sustainable yet profitable manner.

The study also found that governmental support for RJF producers would bring more positivity to the industry than a favourable crude oil price. At the expected USD 60 per barrel and above, some types of RJFs can already be profitable if the same support were provided to RJF producers as was provided to those of biodiesel and bioethanol producers when encouraging policies were drawn up by policy makers worldwide. The RJF-producing industry and governments should avoid the missteps taken during the biodiesel and bioethanol growing phases, by not only pursuing volume and meeting

arbitrary targets but also managing the exploitation of resources more sustainably. Additionally, the use of the 25-indicator RJF sustainability index would complement the pursuit of the more general SDG goals by holistically tracking equitable socio-economic development metrics, which are specifically adapted for the aviation sector.

## Methods

### Feedstock availability

Feedstocks for the production of RJFs are specific to the production methods of OTJ, ETJ and GTJ. For the OTJ method, we selected lipids from the 16 oil crops which include all four major edible feedstocks worldwide such as palm oil, soybean oil, rapeseed oil and sunflower seed oil. The use of vegetable oils as feedstock presents an advantage of having relative uniformity in the lipid properties, where the energy contents and densities are in the small ranges of 36.2–40.1 MJ.kg and 0.9135–0.9620 kg/L, respectively. In the evaluation of sustainability, especially in the light of the "food vs fuel" debate, feedstock availability scenarios cover both export-only and total crop production cases. The export-only feedstock meant that only excess oil from domestic consumption is used. While biofuel productions did exert pressure on the food market, oil only contributed to 12.1% and 9.2% of total calories available for human consumption at global level for developed and developing countries and territories, respectively[19]. Coupled with the growing food wastage issue, the problem to be worried is not of oil diversion for RJFs, but focus should be redirected to resolve the food logistics and distribution issues.

The ETJ feedstocks primarily consist of three sugar-based and nine starch-based crops which will undergo fermentation into ethanol. The biomass-derived ethanol will serve as the starting point for ETJ production methods. Feedstock suitability is more varied due to the large dry matter percentage range of 8.26–89.00 wt% paired with the 16.5–80.0 wt% in sugar or starch content. As ETJ feedstocks are all energy crops, they are afflicted by the same dilemma that of the oil

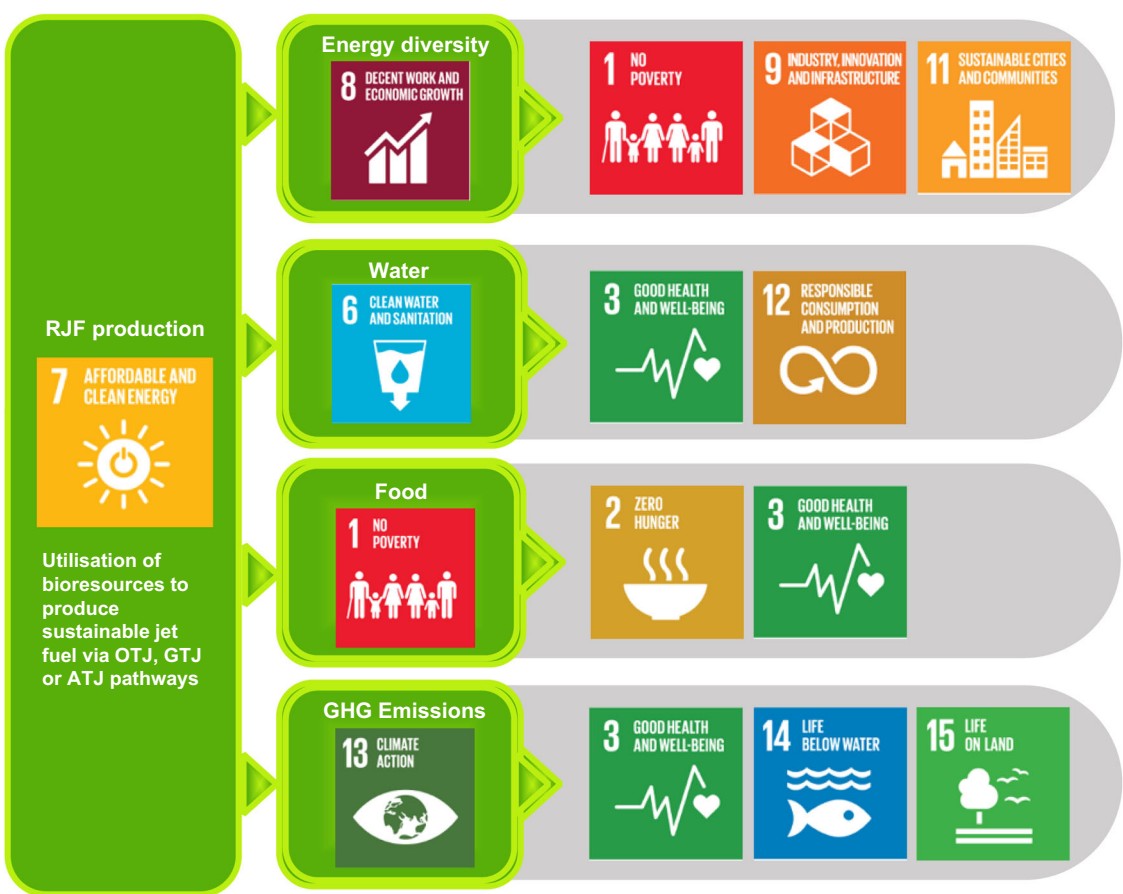

**Fig. 7 | Relevance of renewable jet fuel (RJF) production to the Sustainable Development Goals (SDG)[42].** Mapping of the cascading effect of RJF production on the well-being of society and environment by aligning with the SDG, taking into consideration the spill-over effects of energy diversity, water, food and greenhouse gas emissions.

bearing crops for OTJ. While there is a large body of literature connecting the 2007-08 food price spike to biofuels, there are recent findings that questioned the causal linkages[20]. Kline et al.[21] addressed some earlier misunderstandings and underlying inaccuracies by emphasising that price indices alone are not indicators of food security. In fact, where agricultural land availability is not impeded by the growth of the biofuel sectors, the economic benefits need not be negatively correlated with food security[20]. The eleven GTJ feedstocks are not affected by food security concerns as they are sourced from agricultural wastes which would have otherwise rotted and caused environmental damage. The largest source of GTJ feedstock would come from oil palm fruit which can provide wastes such as palm kernel shell, oil palm frond, oil palm trunk, empty fruit bunch and mesocarp fibre. Refer to Supplementary Dataset 2 for the list of OTJ, ETJ and GTJ feedstocks alongside their key physico-chemical properties, compositions, and economic parameters.

The threshold values for the OTJ, ETJ and GTJ feedstock are a minimum exported quantity of 10, 100 and 100 tonnes per day, respectively. This ensures sufficient feedstock for large-scale production. While the feedstocks for the three RJF production methods do not overlap, they are likely to face competition for usage from the other biofuels such as biodiesel and bioethanol when the prevailing crude oil price is favourable.

### Crude oil price

We used crude oil price as a proxy for jet fuel price, allowing the minimum jet fuel selling price to be determined. While the use of a proxy might be second best compared to directly using RJF prices, it is a trade-off we accepted as jet fuel price is tied closely to the crude oil price. The jet fuel-to-crude oil price ratio has hovered between 0.948 and 1.731. The lowest ratio was an aberration due to the COVID-19 shock in May 2020, when air travel was curtailed, leading to the only time jet fuel is cheaper than crude oil. The highest ratio was caused by the Hurricane Katrina disruption in 2005. Removing outliers, the 20-year variation is consistently close to the average ratio of 1.220. It should be noted that the RJF spot price is not as developed as the benchmark crude oil such as the West Texas Intermediate, Brent Crude, OPEC Reference Basket or the Tapis crude. Thus, the use of crude oil price as proxy is the best available option. In this model, the crude oil price is evaluated for the USD 0-400 per barrel range.

### RJF production technologies

RJF, with the highest technological readiness level, comprises three main conversion pathways, i.e., oil-to-jet (OTJ), gas-to-jet (GTJ) and alcohol-to-jet (ATJ), that can utilise a variety of bioresources for processing. The RJF conversion pathways and suitable feedstock are shown in Supplementary Figure 1. These conversion pathways have been certified by the American Society of Testing Materials (ASTM) standard D7566[8] as drop-in fuel that can be blended with conventional jet fuel, but not all technologies are market-ready for deployment due to high production cost. For the OTJ pathway, hydroprocessed esters and fatty acids (HEFA) are currently the most dominant production method for OTG RJF owing to its flexibility of using first generation feedstocks such as edible oils, or second-generation feedstock such as waste cooking oil or waste lipids. The basic HEFA pathway involves three main processes of deoxygenation, cracking/isomerisation and

distillation, producing straight-chain paraffinic hydrocarbons with high cetane numbers and typically contain no oxygen, sulphur and aromatics[22]. The high oil yield of energy crops such as camelina or edible oil such as rapeseed or palm is the primary reason that drives down the production cost.

The production of ATJ RJF from alcohol can be performed either via the fermentation of sugar with yeast or microbes, or via the combined hydrolyzation-fermentation of starch or lignocellulose. The conversion of lignocellulosic biomass into alcohol requires hydrolysis, followed by fermentation or thermochemical conversion process. The synthesis of ATJ RJF typically undergoes the process of dehydration, oligomerization, hydrogenation and fractionation, using base alcohol such as methanol, ethanol or higher alcohols[23]. From the biomass feedstock, the sugars are fermented to produce iso-butanol or ethanol which will be catalytically dehydrated into isobutylene or ethylene. The oligomerization process creates a carbon chain length suitable for fractionation into fuel components after hydrogenation[24]. ATJ RJF that meets the specification of ASTM standard can be blended with conventional jet fuel up to 50%. The current infrastructure for ethanol production can be retrofitted to produce ATJ RJF, thus driving down the capital investment cost[25].

The GTJ RJF is produced via the Fisher-Tropsch process that converts a mixture of carbon monoxide and hydrogen (synthesis gas) into liquid products of higher molecular weight hydrocarbons[26]. The process was initially developed as a method for making liquid fuels from coal at 400 °C and pressures above 100 bar catalysed with alkalised iron chips[27], which can be extended to other fossil feedstocks such as natural gas and shale gas[28]. The catalytic polymerisation of carbon monoxide occurring during the FT process is accompanied by reaction with hydrogen to make the $CH_2$ methylene units of paraffins before rearranging the molecules via isomerisation to obtain the desired fuel properties[29]. Sustainable biomass was reported to be able to produce RJF via the synthesis gas derived from gasification, which can lead to lower carbon footprint compared to conventional fossil fuels, as the $CO_2$ emitted during the combustion of fuel is offset by the $CO_2$ absorbed during the crop growing process[30]. The challenges of biomass gasification include inconsistent moisture levels, density, energy content, complex lignocellulosic structure and size of biomass, thus the biomass-to-fuel process consists of pre-treatment, gasification, gas conditioning, acid gas removal, FT processing and syncrude refining[31]. The main composition of the FT fuel is normal paraffins and a small fraction of iso-paraffins. Due to the lack of aromatics in the GTJ RJF, the elastomer in the engine was found to shrink, leading to fuel leakage problem[32]. Therefore, blending with conventional jet fuel is required to maintain the level aromatics in aviation turbine fuel, as specified under ASTM standard[8].

## Economic scenarios

The full global biojet fuel sustainability index model allows various combinations in setting the crude oil price (COP) in the USD 0-400 per barrel range (at USD 5 interval), with differing production cost per litre for all three production methods, and two values of subsidy and carbon credit pricing. However, three combinations were evaluated to represent the following scenarios: (i) High crude oil price with subsidies and market measures: COP = USD135/barrel, production costs for OTJ = -USD 0.6378/litre, ETJ = USD 0.6334, GTJ = -USD 0.1824 (ii) Moderate crude oil price with subsidies and market measures: COP = USD75/barrel, production costs for OTJ = -USD 0.6378/litre, ETJ = USD 0.6334, GTJ = -USD 0.1824 (iii) Moderate crude oil price with subsidies but no market measures: COP = USD75/barrel, production costs for OTJ = -USD 0.2415/litre, ETJ = USD 1.0296, GTJ = -USD 0.2139. See Supplementary Dataset 3 for the permutations of economic scenarios for this study. The first scenario is selected as the baseline case, as per discussed in the main text. We have decided to keep government subsidies in all scenarios, as both the biodiesel and bioethanol

industries around the world were incentivized by fiscal measures in the nascent stages of growth. It does not matter whether the subsidy comes in the form of pioneering benefits, cash payments or tax reductions.

## Country and territory ranking

All the 154 countries and territories evaluated are ranked according to the aggregate scores from all four domains of the Global Biojet Fuel Sustainability Index, namely energy security, water security, food security and governance. They cover the entire EWF nexus, while also factoring in government effectiveness in ramping up RJF production under market conditions. See Supplementary Dataset 4 for the breakdown of the indicator parameters, data required and weightage for the four domains, nine themes, 16 sub-themes and 25 indicators within the Global Biojet Fuel Sustainability Index. The maximum possible index score is 100.

## Energy security

The energy domain is the largest contributor to the sustainability index with a 40% weightage. This domain was deliberately selected as the most influential as RJF is supposed to solve the energy problem linked to SDG 7, while minimising the stress on the finite resources of water (SDG 6) and not aggravating food security (SDG 2). We divided the security into three themes, such as production (21%), energy use (9%) and environmental emissions (10%). Production covers the present-day RJF production, potential RJF production quantity and source diversification to ascertain a country's production capacity. The index favours countries and territories with existing RJF industries. Environmental emissions factors in the CO, $NO_x$, UHC and soot emissions associated with the transition to RJF.

Energy use takes into account the impact of integrating RJF into the energy mix and its effect on import dependency. While RJF is likely to diversify the energy mix of a country or territory, converting a large portion of energy to biomass-based energy might do the opposite. The Herfindahl-Hirschman index was employed to look at the energy diversity once all eligible feedstock is converted into RJF. On the surface, the transition of imported fossil-based jet fuel to locally produced RJF has no drawbacks. However, a biomass-dominant system, especially in a monocultured agricultural setting, is susceptible to diseases, pests and devastating effects on the natural ecosystem. This theme attempts to balance the benefits of RJF while considering the other components within the EWF nexus. We used datasets from UNdata[33] (RJF production), USDA[34] (biofuels production), FAOSTAT[35] (crop trades, land, fertiliser usage), EIA[36] (jet fuel consumption), IEA[37] (energy sources) and UN Population[38] (demographics) for this domain. See Supplementary Dataset 5 for the list of datasets used.

## Water security

We assigned a weight of 25% for water security, while splitting it into the two themes of water stress (5%) and agriculture (20%). Datasets from FAO-AQUASTAT[39] (agricultural water withdrawal, renewable water resource and precipitation) and CWASI[40] (water footprint) were utilised. Water stress evaluates how the hypothetical transition to RJF would affect the baseline water stress index and also the water withdrawal per capita. We also looked at how the planting of crops associated with RJF feedstocks will affect the green, blue and grey water scarcity indices and agricultural dependency. Green water refers to the rainwater used, blue water denotes the irrigation water requirement, and grey water is the amount of fresh water utilised in diluting pollution. Each feedstock has its own water requirement, in turn impacting the water stress levels.

## Food security

This domain is weighted at 25% of the total sustainability index allocation, with three themes of availability (13%), access (7%) and stability

(5%). Availability looks at ready disposal of arable land, crop yield (and its volatility over a 15-year period) and adequacy of food supply to the population. We consider food access to be economics-oriented, as countries and territories are likely to be able to divert excess food for RJF production only if food deficit and undernourishment are non-issues.

In addition to the volatility of crude oil price, feedstock price is another component with high price fluctuations. Such fluctuations make it untenable to set up business models for RJF productions, unless there are financial guarantees such as government subsidies. As such, domestic crop price volatility over a 15-year period is used to determine the adverse effects of price shocks. We exclusively used FAOSTAT[35] datasets to obtain the yearly (undernourishment, crop yield and production), 3-year average (dietary requirements) and 15-year average (producer price, crop yield and production) values.

## Governance

A country or territory could hold an advantageous position within the EWF nexus, yet transitioning away from fossil jet fuels demands strong governance. The six key dimensions of governance as identified by the World Bank such as accountability, political stability, effectiveness, regulatory quality, rule of law and corruption control are all equally weighted to form 10% of the sustainability index scores. For this, we used the World Governance Indicators dataset as obtained from the World Bank[41].

## Data availability

The full set of equations for the model can be found in Supplementary Information item no. 7. The processed data that extends upon the findings of this study are provided in Supplementary Dataset, including source data used to plot the figures in Supplementary Dataset 6. An interactive website showing additional economic scenarios can be found at indexsustainability.com. Other information is available from the corresponding author upon request.

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

## Acknowledgements
This work was supported by the research fund of International Excellent Young Scientists (52250610220) from the National Natural Science Foundation of China (NSFC) awarded to C.T. Chong.

## Author contributions
C.T.C. modelled, analysed, and validated the result, J.-H.N. conceptualised, modelled, and analysed the result.

## Competing interests
The authors declare no competing interests.
