## [Peer review file · Nature Communications]

REVIEWER COMMENTS

Reviewer #1 (Remarks to the Author):

I have reviewed the manuscript Limitations to sustainable renewable jet fuels production attributed to cost than energy-water-food resource availability. Next, I share my comments.

1. In the introduction, please include the revision of the related literature in order to clarify the contribution of the study.
2. Information related to the organization of the manuscript must be provided.
3. Based on what criteria the weights for each aspect were chosen?
4. The analysis must be complemented considering the RSB-CORSIA standard.
5. Why are not included in the study waste raw materials?
6. How these values were chosen: The threshold values for the OTJ, ETJ and GTJ feedstock for each country is a 393 minimum exported quantity of 10, 100 and 100 tonnes per day

Reviewer #2 (Remarks to the Author):

The work focuses on the evaluation of the sustainability of renewable jet fuels (RJFs) from a global perspective. A total of 155 countries have been chosen to assess the potential of RJF production, taking into account the potential feedstock and production pathways. The authors have developed a methodology to rank the sustainability index of RJF production, using the criteria of four domains, i.e. food, water, energy and governance. The biojet fuel sustainability index scores of the countries are derived based on 33 indicators. From there, aggregate score for each country is obtained and then ranked.

Then, the assessed sustainability criteria are elaborated with in-depth analysis from the perspectives of EWF - energy replaceability, water footprint, feedstock availability and food security. Substantial effort has been put in to quantify each of the metrics at country level. Next, an economic analysis was conducted to assess the profitability of renewable jet fuels based on different scenarios of crude oil prices and various subsidy levels. From there, the main limiting factors for the development of renewable jet fuel are identified for each of the countries. Finally, the authors assessed the sustainability of renewable jet fuel are also aligned to the UN Sustainable Development Goals (SDG) metrics.

Overall, this work is an insightful work regarding the quantification of sustainability of renewable jet fuel from the global perspective. The methodology and analysis used are sound, supported by an extensive set of data that is up-to-date. This work provides the insights for the decarbonization of aviation sector. Here are some comments on the strengths and suggestions for improvement.

The strengths of the paper are:

1. The coverage of the work is extensive with a global perspective. With 155 countries, almost all countries in the world with available data are assessed. The data used are derived from established global database such as FAO, World Bank etc, thus, the ranking for the sustainability of the countries can be considered transparent, accurate and fair. A major strength from the analysis is that it leads to the identification of the major limiting factors to RJF production, which is informative and conclusive.
2. RJF is a hot topic nowadays in the area of decarbonization but there is considerable lack of study on it. There are some papers regarding the CO₂ footprint assessment of biojet fuels in the Nature series, but there is a lack of a holistic sustainability assessment of this area. This work is timely to fill the gap, and compliments the studies of GHG reduction of jet fuel production.
3. The model presented is very detail, taking into account the important factors of EWF nexus, subsidies and crude oil price. The complex interaction of the factors is taken into account in their model, which is more in-depth than some papers that only consider the EWF or localized economic models.
4. The quantitative study is aligned with the SDG metrics, which is useful to present the impacts of RJF from another angle. The mapping of the spill-over effects from RJF production supports the conclusion from the analytical model and is highly relevant to sustainable development.

Some suggestions to improve the paper:

1. It is important to keep the paper up-to-date. The author made the effort to tie in with the current Ukraine-Russian conflict and fluctuating oil price. There are other current issues such as the European/global energy crisis which was partially caused by the supply chain crisis, OPEC supply restriction and climate abnormalities. Would such current issue impact the model and result, and should the model be updated?

2. The figures presented are compact and informative, but the downside is the lack of clarity as the wordings seem small, exp Fig. 1, box 1. If possible, please enhance the legibility of the figure. Figure 2 and 3 are interesting, but the interpretation of the figure is not straightforward. Perhaps the author can elaborate using 1 or 2 distinct examples to aid the understanding to the reader?

3. In the methodology section, the author assumed the RJF production technology to consist of only OTJ, ETJ and GTJ. How about other new emerging methods such as direct sugar conversion? Are they methods (new) that do not fall under the category might be more effective?

4. Another issue is feedstock. Certain big countries may have multiple feedstock, apart from some main crops. Did the author consider the different types of feedstock in one countries? What about the different generations of feedstock used to produce the RJF? Or what are the criteria for feedstock selection in the model?

REVIEWER COMMENTS

Reviewer #1 (Remarks to the Author):

I have reviewed the manuscript Limitations to sustainable renewable jet fuels production attributed to cost than energy-water-food resource availability. Next, I share my comments.

1. In the introduction, please include the revision of the related literature in order to clarify the contribution of the study.

Reply: The authors have added contribution of the study towards the end of the Introduction section (page 4):

“With this, the study aims to advocate for the use of a sustainability index for RJF to complement the pursuit of the SDGs, while enabling a more comprehensive assessment of RJF sustainability tailored to the aviation sector. This will provide insights to support the aviation industry’s efforts to meet the global sustainability targets.”

2. Information related to the organization of the manuscript must be provided.

Reply: The authors have added additional information related to the organization of the manuscript at the end of the Introduction section (page 4):

“Here, we report in the following order covering the global biojet fuel sustainability index by ranking the countries on their sustainability measures, followed by the three key thrusts of the nexus, such as energy replaceability, water footprint, and food along with feedstock. Then, the economics aspects on the profitability price point and the overall limiting factors within the resource nexus are discussed. Then, the sustainability of RJF is elaborated from an SDG perspective.”

3. Based on what criteria the weights for each aspect were chosen?

Reply: The ‘Sustainability index indicators’ was developed to have four main categories, covering the securities within the energy-water-food (EWF) nexus and governance. From the four categories, a weighted sum model (WSM) was applied to evaluate the sustainability scores of each country, while also factoring in the relative weight of importance of the criteria.

There are two layers of weights that were assigned, namely:

- 1) **Categorical weights, CW** (Energy security – 40%, Water security – 25%, Food security – 25%, Governance 10%)
- 2) **Indicator weights, IW** (there are 25 indicators across the four categories)

The CWs are the sum all IWs within the category. Despite being the sum of the IWs, the CWs were first assigned to set the boundaries for each category. As the sustainability index is for RJFs, energy is arbitrarily given the greatest weight of 40%. This is followed by the equal weightage (25% each) given to the two other securities of water and food. The remaining 10% is assigned to governance.

Once the CWs were assigned, the IWs within the category will have an upper limit where weights can be assigned. From there, weights for the IWs were assigned by the authors based on its expected effects

of the respective categories. This leads to some heavily weighted indicators like the current RJF production which carries 10% weight, as it represents the steps already taken by countries to move towards a more sustainable aviation sector. On the other hand, there are also lightly weighted indicators like feedstock source diversity at 1%, as the range of feedstock is arguably less important than the quantity of it.

4. The analysis must be complemented considering the RSB-CORSIA standard.

Reply: The authors acknowledge the reviewer’s comment on the RSB-CORSIA certification standard, and is of the opinion that the international sustainability certification does provide a framework to verify the sustainability and GHGs reduction of RJFs (also more precisely called Sustainable Aviation Fuel (SAF) under the programme).

However, the scope of the RSB-CORSIA programme is limited as compared to the scope that the sustainability index model intends to cover in this study. A quick comparison can be shown in the table below:

Type	RSB-CORSIA certification programme	Present Sustainability Index model
Objective scope	 • Feedstock production • Land usage • Production method • GHG emissions 	25 indicators covering the energy-water-food nexus and governance, which covers the broad categories of:  • Feedstock production • Energy use • GHG emissions • Water stress • Water scarcity • Land usage • Food access • Food volatility While the scope of human rights and governance-related items are covered under ‘Governance’.
Subjective scopes	 • Traceability • Human rights • Biodiversity • Sustainable livelihood 	
Focus	Carbon offsetting and reduction	Holistic view of sustainability from a nexus-governance perspective.

Additionally, the present model aims to quantify and rank the potential sustainability from a total resource point-of-view. Thus, applying what amounts to be a RSB-CORSIA filter in the analysis will be limiting, as the RSB-CORSIA approved RJFs will only be a very small subset of the potential RJF covered in this study. In fact, there are only 32 countries as members (mostly developed countries) under the RSB-CORSIA [1]. This goes against the intended scope of this study to cover as many countries as possible.

Another weakness of using RSB-CORSIA in the present analysis would be to reduce the model to what is already presently compliant rather than what could potentially be included under the

programme in the future. This is without considering the often-changing certification type, validity dates and certification status.

[1] RSB, CORSIA in the RSB certification system. December 2020. [Access: 24 March 2023] <https://rsb.org/wp-content/uploads/2020/11/CORSIA-in-the-RSB-System-Overview.pdf>

5. Why are not included in the study waste raw materials?

Reply: Study of waste raw material was initially considered in the model, but was eventually omitted from the final model. The omission was intentional as data for waste as raw material are scarce, poorly defined and typically non-uniform from a data point of view. For example, unlike food production data where the key parameters are well-defined and uniform globally, the definition of waste for each country differs greatly.

The gold standard for data on waste that is most useful for conversion into RJF would be municipal solid waste (MSW). Most countries do not segregate food waste from household waste with other non-domestic and industrial wastes. This is compounded by the lack of breakdown of the MSW type by their categories such as food, organics, paper, plastic, metal, glass, inert materials, and textiles. As such, it will be difficult to quantify the actual availability of waste feedstock for RJF production. The authors took a stand to prevent spurious data from being used in this data-rich model.

For more uniformed data, the data are typically about a decade old (roughly between 2008-2012). Compounding to the problem is most non-OECD countries (in particular from Africa, Pacific islands and Caribbean) do not even have such data at national level. This in turn became a two-fold problem of:

- i) Equity
- ii) Gatekeeping

The model is meant to quantify and qualify the potential for producing RJF in a fair and equitable manner, as such we chose not to omit countries due to the lack of waste data, but instead omit the use of waste as raw materials. Shall the gatekeeping measure of using waste data to omit countries be utilised, then the model would not have as many countries that meet the data criteria for inclusion in this model.

The inclusion of waste as raw material will be revisited as waste data becomes more ubiquitous for all countries. When that happens, it will be useful to also factor in economic calculations such as 'Avoided Landfill Emissions Credits (LEC) and Recycling Emissions Credit (REC) for sustainable aviation fuels derived from Municipal Solid Waste (MSW) [2]. These will be more suitable when the model expands to include carbon offsetting, rather than gauging potential and measuring sustainability as the model sets out to presently do.

[2] ICAO, 'CORSIA Methodology for Calculating Actual Life Cycle Emissions Values', June 2022, [Access: 24 March 2023], https://www.icao.int/environmental-protection/CORSIA/Documents/CORSIA_Eligible_Fuels/ICAO%20document%2007%20-%20Methodology%20for%20Actual%20Life%20Cycle%20Emissions%20-%20June%202022.pdf

6. How these values were chosen: The threshold values for the OTJ, ETJ and GTJ feedstock for each country is a 393 minimum exported quantity of 10, 100 and 100 tonnes per day

Reply: The minimum threshold values for the feedstock for each country is based Johnston and Holloway's [3] volume throughput estimate of an efficient large-scale, continuous flow oil feedstock reactor. The rationale for the threshold value is to ensure that there are cost-effective industries that can be formed while ensuring that feedstock are available within the country (further tying in to the measure of energy security). For oil-based feedstock, the amount was estimated to be 1 million L of production volume, which after taking into consideration three factors (*day to annual conversion, mass to litres conversion and efficiency of process*), led to a rounded off value of 10 tonnes per day. The same rationale was used for ETJ and GTJ which eventually amounted to 100 tonnes per day.

[3] Johnston M, Hollloway T, 'A Global Comparison of National Biodiesel Production Potentials', *Environ. Sci. Technol.*, 2007, 41(23), pp 7967-7973.

Reviewer #2 (Remarks to the Author):

The work focuses on the evaluation of the sustainability of renewable jet fuels (RJFs) from a global perspective. A total of 155 countries have been chosen to assess the potential of RJF production, taking into account the potential feedstock and production pathways. The authors have developed a methodology to rank the sustainability index of RJF production, using the criteria of four domains, i.e. food, water, energy and governance. The biojet fuel sustainability index scores of the countries are derived based on 33 indicators. From there, aggregate score for each country is obtained and then ranked.

Then, the assessed sustainability criteria are elaborated with in-depth analysis from the perspectives of EWF - energy replaceability, water footprint, feedstock availability and food security. Substantial effort has been put in to quantify each of the metrics at country level. Next, an economic analysis was conducted to assess the profitability of renewable jet fuels based on different scenarios of crude oil prices and various subsidy levels. From there, the main limiting factors for the development of renewable jet fuel are identified for each of the countries. Finally, the authors assessed the sustainability of renewable jet fuel are also aligned to the UN Sustainable Development Goals (SDG) metrics.

Overall, this work is an insightful work regarding the quantification of sustainability of renewable jet fuel from the global perspective. The methodology and analysis used are sound, supported by an extensive set of data that is up-to-date. This work provides the insights for the decarbonization of aviation sector. Here are some comments on the strengths and suggestions for improvement.

Reply: The authors agree with the assessment of the paper, and it accurately reflects the ideas espoused in this study.

The strengths of the paper are:

1. The coverage of the work is extensive with a global perspective. With 155 countries, almost all countries in the world with available data are assessed. The data used are derived from established global database such as FAO, World Bank etc, thus, the ranking for the sustainability of the countries can be considered transparent, accurate and fair. A major strength from the analysis is that it leads to the identification of the major limiting factors to RJF production, which is informative and conclusive.

Reply: The authors thank the reviewer for the comments on the strength. Indeed, one of the methodologies adopted in this study is the formulation of a data-driven model using only dataset from

reputable sources. Thus, data used are from well-known and easily accessed (*and well-archived*) portals such as FAOstat, World Bank, UNDP and others. Additionally, the technical ‘constants’ (*such as the physico-chemical properties and chemical conversion processes*) were referenced from well-known studies. The ability of the model to establish the limiting factors for each country is one of the key strength of this model, alongside the ability to rate (*and rank*) countries on their sustainability measures from the energy-water-food nexus plus governance aspects.

2. RJF is a hot topic nowadays in the area of decarbonization but there is considerable lack of study on it. There are some papers regarding the CO₂ footprint assessment of biojet fuels in the Nature series, but there is a lack of a holistic sustainability assessment of this area. This work is timely to fill the gap, and compliments the studies of GHG reduction of jet fuel production.

Reply: The authors thank the reviewer for the astute insights. Indeed, renewable jet fuel (RJF) is a trending topic in research especially in the post-pandemic world as the aviation industry is expected to maintain its relentless growth rate. In this case, the growth centre is expected to centre around South East Asia and Latin America, with routes in/out of them increasing. As such, there is a need for a work which focuses on regions outside of North American, Europe and East Asia. In this case, the authors decided to do a global study to cover all regions. Additionally, the authors also felt that most studies involving sustainability only looked at life cycle assessments, which is holistic from a carbon perspective, but insufficient from an energy-water-food nexus and governance perspective. In fact, even the CORSIA eligible fuel (CEF) used to certify the sustainability qualities of a fuel is embroiled in some level of controversy over the direct and indirect land usage measures. Thus, the author believes that this work is timely and complements the other sustainability-related studies for RJF.

3. The model presented is very detail, taking into account the important factors of EWF nexus, subsidies and crude oil price. The complex interaction of the factors is taken into account in their model, which is more in-depth than some papers that only consider the EWF or localized economic models.

Reply: The authors again thank the reviewer for the statement. The model is indeed one of the most comprehensive model of its type, with 13 interlinked sections (as shown in the 128-page calculation manual, which was submitted as “Supplementary Document 1 – Equations”).

The model was also deliberately designed to distinguish itself from localised economic models (e.g. case studies), as the focus of the paper is to provide a macro view of RJF with respect to sustainability measures. Furthermore, the study also goes one step further than EWF by also factoring in governance, as the governmental will and current regulations are also a co-enabler to the limiting factors of energy, water and food.

4. The quantitative study is aligned with the SDG metrics, which is useful to present the impacts of RJF from another angle. The mapping of the spill-over effects from RJF production supports the conclusion from the analytical model and is highly relevant to sustainable development.

Reply: Most RJF studies would intuitively link themselves to SDG 7 (Affordable and Clean Energy). In this study, we consider the intertwined nature of the EWF nexus, and related them to what should also be the intertwined goals of the SDG. Here, we linked up the EWF nexus to 12 of the SDGs, as shown in Fig 6 of the manuscript (Line 320, pg15).

Some suggestions to improve the paper:

1. It is important to keep the paper up-to-date. The author made the effort to tie in with the current Ukraine-Russian conflict and fluctuating oil price. There are other current issues such as the European/global energy crisis which was partially caused by the supply chain crisis, OPEC supply restriction and climate abnormalities. Would such current issue impact the model and result, and should the model be updated?

Reply: The authors agree with the suggestions by the reviewer. The paper remains up to date as the economic scenarios highlighted in the paper still cover the range for the fluctuating oil prices (*i.e.* USD 75-135/barrel for crude oil price). In fact, the model (as shown in Fig 4 of the manuscript) covers the crude oil price range of USD 0-400/barrel, which is likely to cover all possible economic scenarios. The highest record for crude oil price is slightly under USD 150/barrel in July 2008 [4-5]. Nan et. al. [5] supported the point that the global pandemic of COVID-19 and other geopolitical events mentioned by the reviewers did cause major price fluctuations, but they still remain within the range of the model.

[4] Gang Wu, Yue-Jun Zhang, 'Does China factor matter? An econometric analysis of international crude oil prices', 2014, Energy Policy 72, pp 78-86.

[5] Yu Nan, Renjin Sun, Zhao Zhen, Chu Fangjing, 'Measurement of international crude oil price cyclical fluctuations and correlation with the world economic cyclical changes', 2022, Energy 260, p.124946.

2. The figures presented are compact and informative, but the downside is the lack of clarity as the wordings seem small, exp Fig. 1, box 1. If possible, please enhance the legibility of the figure. Figure 2 and 3 are interesting, but the interpretation of the figure is not straightforward. Perhaps the author can elaborate using 1 or 2 distinct examples to aid the understanding to the reader?

Reply: The authors agree with the reviewer on the need to improve the legibility of the figure in Box 1, and provide distinct examples for Figures 2 and 3 to aid the understanding of readers. The following are the changes made or highlighted in revised text:

- 1) **Page 8:** The image contained within Box 1 has been rearranged to improve legibility of the in-image text.
- 2) **Page 9:** It is emphasised that some of the largest jet fuel consumers (such as USA, China, India, Russia and Brazil) are also the ones able to potentially produce profitable biojet fuel, and all of them are able to exceed the maximum RJF blending level of 50%. Countries that could not achieve the maximum RJF blend levels (*i.e.* those positioned below the red line in Figure 2) but presently consume high quantities of jet fuel are also named.
- 3) **Pages 11-12:** The significance of the quadrants are made clearer to the readers. Added a part about the Asian continent being in Quadrant II, which is the region with low blending ratio potential coupled with water stress.

3. In the methodology section, the author assumed the RJF production technology to consist of only OTJ, ETJ and GTJ. How about other new emerging methods such as direct sugar conversion? Are they methods (new) that do not fall under the category might be more effective?

Reply: The OTJ, ETJ and GTJ methods (or more precisely the family of methods that is categorised by the feedstock they use) do also cover emerging methods such as direct sugar-to-hydrocarbon

conversion (DSHC). DSHC would be categorised under ETJ for the purpose of this model as it falls under the sugar-alcohol feedstock group.

A mapping of the ASTM D7566 approved SAF conversion method to the family of production methods used in this model is shown below in Table 1:

Table 1: Mapping of the ASTM D7566 approved SAF conversion methods to the production methods used in the model (information on the annexes adapted from [6]).

ASTM D7566 Annexes	Conversion process	Possible feedstock	Mapping to OTJ/ETJ/GTJ in this model
Annex 1	Fischer-Tropsch (FT)	Coal, natural gas, biomass	GTJ
Annex 2	Hydroprocessed esters and fatty acids (HEFA)	Bio-oils, animal fats, recycle oils	OTJ
Annex 3	Synthesized iso-paraffins (SIP)	Biomass used for sugar production	ETJ
Annex 4	Synthesized kerosene with aromatics derived by alkylation (FT-SKA)	Coal, natural gas, biomass	GTJ
Annex 5	Alcohol-to-jet (ATJ-SPK)	Biomass from ethanol/isobutanol production	ETJ
Annex 6	Catalytic hydrothermolysis jet fuel (CHJ)	Triglycerides	OTJ
Annex 7	Synthesized paraffinic kerosene from hydrocarbon – hydroprocessed esters and fatty acids (HC-HEFA-SPK)	Algae	OTJ

While the authors are aware of the growing number of approved conversion methods, the O/E/GTJ classification are used to future proof the model, and to prevent excessive fracturing and double-counting of the conversion methods within the model.

[6] ICAO, ‘Conversion Processes’, [Accessed Date: 24 March 2023], <https://www.icao.int/environmental-protection/GFAAF/Pages/Conversion-processes.aspx>

4. Another issue is feedstock. Certain big countries may have multiple feedstock, apart from some main crops. Did the author consider the different types of feedstock in one countries? What about the different generations of feedstock used to produce the RJF? Or what are the criteria for feedstock selection in the model?

Reply: Yes, the model factored in multiple feedstocks for all countries, as long as the feedstock quantity exceeded the threshold. In other words, countries are not limited only to the dominant

feedstock/crops. The model covers existing first and second generation feedstocks, with the breakdown being 16 oil crops for OTJ, 3 sugar-based and 9 starch-based for ETJ and 11 GTJ feedstocks.

The criterion for inclusion into the model is their suitability for conversion into RJF and its existing availability in large quantities. As such, third generation (algae, yeast, bacteria) and fourth generation (genetically enhanced feedstock) were omitted from this model as they have not yet reached the required critical mass. The authors are open to their inclusion in future updates of this work.

REVIEWERS' COMMENTS

Reviewer #1 (Remarks to the Author):

I have reviewed the new version of the manuscript. All my previous concerns have been addressed.

Reviewer #2 (Remarks to the Author):

The authors have incorporated comments from the first round of review. My concerns from my previous review have been addressed. I would recommend the paper to be accepted for publication.